# MORE OR LESS: WHEN AND HOW TO BUILD CONVOLUTIONAL NEURAL NETWORK ENSEMBLES

**Abdul Wasay**
Harvard University
awasay@seas.harvard.edu

**Stratos Idreos**
Harvard University
stratos@seas.harvard.edu

## ABSTRACT

Convolutional neural networks are utilized to solve increasingly more complex problems and with more data. As a result, researchers and practitioners seek to scale the representational power of such models by adding more parameters. However, increasing parameters requires additional critical resources in terms of memory and compute, leading to increased training and inference cost. Thus a consistent challenge is to obtain as high as possible accuracy within a parameter budget. As neural network designers navigate this complex landscape, they are guided by conventional wisdom that is informed from past empirical studies. We identify a critical part of this design space that is not well-understood: How to decide between the alternatives of expanding a single convolutional network model or increasing the number of networks in the form of an ensemble. We study this question in detail across various network architectures and data sets. We build an extensive experimental framework that captures numerous angles of the possible design space in terms of how a new set of parameters can be used in a model. We consider a holistic set of metrics such as training time, inference time, and memory usage. The framework provides a robust assessment by making sure it controls for the number of parameters. Contrary to conventional wisdom, we show that when we perform a holistic and robust assessment, we uncover a wide design space, where ensembles provide better accuracy, train faster, and deploy at speed comparable to single convolutional networks with the same total number of parameters.

## 1 INTRODUCTION

**Scaling capacity of deep learning models.** Convolutional neural network models are becoming as accurate as humans on perceptual tasks. They are now used in numerous and diverse applications such as drug discovery, data compression, and automating gameplay. These models increasingly grow in size with more parameters and layers, driven by two major trends. First, there is a continuous rise in data complexity and sizes in many applications (Shazeer et al., 2017). Second, there is an increasing need for higher accuracy as models are utilized in more critical applications – such as self-driving cars and medical diagnosis (Grzywaczewski, 2017). This effect is especially pronounced in computer vision and natural language processing: Model sizes are three orders of magnitude larger than they were just three years ago (Sanh et al., 2019).

With bigger model sizes, the time, computation, and memory needed to train and deploy such models also increase. Thus, it is a consistent challenge to design models that maximize accuracy while remaining practical with respect to the resources they need (Lee et al., 2015; Huang et al., 2017b). In this paper, we study the following question: Given a number of parameters (neurons), how to design a convolutional neural network to optimize holistically for accuracy, training cost, and inference cost?

**The holistic design space is very complex.** Designers of convolutional neural network models navigate a complex design landscape to address this question: First, they need to decide on network architecture. Then, they have to consider whether to use a single network or build an ensemble model with multiple networks. Additionally, they have to decide how many neural networks to use and their individual designs, i.e., the depth, width, and number of networks in their model. Modern

applications with diverse requirements further complicate these decisions as what is desirable varies. Facebook, for instance, requires convolutional neural network models that strike specific tradeoffs between accuracy and inference time across 250 different types of smartphones (Wu et al., 2019). As a result, not just accuracy but a diversity of metrics – such as inference time and memory usage – inform whether a model gets used (Sze et al., 2017b).

**Scattered conventional wisdom.** There exist bits and pieces of scattered conventional wisdom to guide a neural network designer. These take the form of various empirical studies that demonstrate how depth and width in a single neural network model relate to certain metrics such as accuracy. First, it is generally known that deeper and wider networks can improve accuracy. In fact, recent convolutional architectures – such as ResNets and DenseNets – are designed precisely to enable this outcome (He et al., 2016; Huang et al., 2017b;a). The caveat with beefing up a neural network is that accuracy runs into diminishing returns as we continue to add more layers or widen existing ones (Coates et al., 2011; Dauphin and Bengio, 2013). On the other hand, increasing the number of networks in the model, i.e., building ensembles, is considered a relatively robust but expensive approach to improve accuracy as ensemble models train and deploy $k$ networks instead of one (Russakovsky et al., 2015; Wasay et al., 2020). The consensus is to use ensembles when the goal is to achieve high accuracy without much regard to training cost, inference time, and memory usage, e.g., competitions such as COCO and ImageNet (Lee et al., 2015; Russakovsky et al., 2015; Huang et al., 2017a; Ju et al., 2017). All these studies, however, exist in silos. Any form of cross-comparison is impossible as they use different data sets, network architectures, and hardware.

**Lack of a robust and holistic assessment.** Most past studies operate within the confines of a single convolutional network and do not consider the dimension of ensemble models. Those that compare with ensembles mostly do so unfairly comparing ensembles with $k$ networks against a model that contains only one such network (Lee et al., 2015; Russakovsky et al., 2015; Huang et al., 2017a; Ju et al., 2017). There are recent studies that make this comparison under a fixed parameter budget (Chirkova et al., 2020; Kondratyuk et al., 2020). However, these studies consider only the metric of generalization accuracy and explore a very small part of the design space – two different classes of convolutional architectures with a single depth.

A holistic analysis needs to include resource-related metrics such as training time, inference cost, and memory usage. All these metrics are critical for practical applications (Sze et al., 2017a; Wu et al., 2019). Furthermore, to provide reliable guidance to a model designer, a robust comparison needs to consider a range of architectures and model sizes with various depth and width configurations. This is critical, especially because varying just the width of convolutional networks in isolation, as done by recent studies (Chirkova et al., 2020; Kondratyuk et al., 2020), is known to be far less effective to improve accuracy (Eigen et al., 2013; Ba and Caruana, 2014).

**Single networks vs. ensembles.** In this paper, we bridge the gap in the understanding of the design space by providing answers to the following questions. *Given specific requirements in terms of accuracy, training time, and inference time, should we train and deploy a convolutional model with a single network or one that contains an ensemble of networks? How should we design the networks within an ensemble? As these constraints and requirements evolve, should we switch between these alternatives, why, and when?*

**Method.** We introduce the following methodology to map the design space accurately. Since there is no robust theoretical framework to consistently analyze the design space and the complex interactions among its many parameters and metrics, we develop a detailed and extensive experimental framework to isolate the impact of the critical design knobs: (i) depth, (ii) width, and (iii) number of networks, on all relevant metrics: (i) accuracy, (ii) training time, (iii) inference time, and (iv) memory usage. Crucially the number of parameters is a control knob in our framework, and we only compare alternatives under the same parameter budget. To establish the robustness of our findings, we experiment across various architectures, data complexities, and classification tasks. We present and analyze data amounting to over one year of GPU run time. We also explain trends breaking down metrics into their constituents when necessary.

**Results: The Ensemble Switchover Threshold (EST).** (i) Contrary to conventional wisdom, we show that when we make a holistic and robust comparison between single convolutional networks

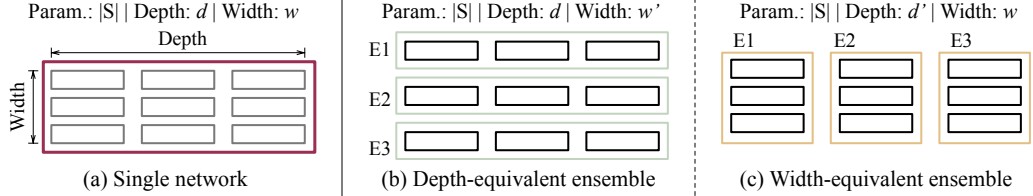

Figure 1: We explore a design space consisting of three design classes: (a) Single convolutional network models, (b) Depth-equivalent ensembles, and (c) Width-equivalent ensembles. The two ensemble design classes are created by distributing either the width factor or the depth corresponding to the single network amongst the ensemble networks while keeping the other factor fixed.

and ensembles of networks, we discover a vast design space where ensembles provide not just better overall accuracy but also train faster compared to a single network. (ii) Specifically, we uncover the Ensemble Switchover Threshold (EST). This is the amount of resources (measured in terms of the number of parameters and training epochs) beyond which ensembles provide superior generalization accuracy to a single model. (iii) We show that EST occurs consistently across numerous data sets and architectures. (iv) We demonstrate that the number of networks in an ensemble and their individual designs determine the EST. (v) Ensembles can also provide comparable inference times for a considerable part of the design space. (vi) We also show that ensembles require significantly less memory to train for the same number of parameters. (vii) Finally, we make available a superset of our results for visual exploration and help with model design at: `daslab.seas.harvard.edu/more-or-less.`,

## 2 FRAMEWORK: DESIGN SPACE

The design space we explore consists of single convolutional neural network models and two classes of architecturally-homogenous ensembles. These ensemble classes help isolate the effect of the two design knobs – depth and width – on the quality and cost of an ensemble design. We first describe how we ensure a robust comparison of alternative model designs and then explain the degrees of freedom we explore.

**Establishing grounds for fair ranking.** A key element of our framework is that the possible model designs are compared to each other only under equivalent resources. We ensure this by only comparing designs that have the same number of parameters. This comparison allows us to separate the quality of a design from the amount of resources given to it. Another way to think about this is that given a parameter budget, we can investigate how the three design classes rank for all relevant metrics (training and inference time, accuracy, and memory usage).

We fix the number of parameters because of its two distinctive properties over other metrics (that we could have fixed), such as training time, inference time, or accuracy: First, the number of parameters of a network is directly proportional to all other resource-related metrics (Jain et al., 2020; Wasay et al., 2020). Second, the number of parameters is independent of the hardware or the software platform used and can be computed exactly from a network specification.

**The single network versus ensemble design space.** Our design space considers a convolutional neural network architecture $S^{(w,d)}$ from a class of neural network architectures $C$. $S^{(w,d)}$ has width factor $w$, depth $d$, and number of parameters $|S|$. Similarly an ensemble is described as $E = \{E_1 \ldots E_k\}$. Ensembles are architecturally-homogenous i.e., all ensemble networks $E_1 \ldots E_k$ have the same architecture and each network has $|E|/k$ parameters. When we compare a single network $S^{(w,d)}$ from $C$ with an ensemble $E$ we ensure that $E_1 \ldots E_k \in C$ and $|E_1| + \ldots + |E_k| = |S|$.

The reason why we restrict the design space to homogenous ensembles is to reduce the otherwise intractably large space[1] of all possible ensembles given a single network to a size that we can

---

[1]Given a single network with $|S|$ number of parameters, there are $\left\{ {|S| \atop k} \right\}$ (Stirling number of the second kind) as many ways of forming ensembles of size $k$. This number grows at a similar rate to exponential polynomials, (Boyadzhiev, 2009) e.g., $\left\{ {100 \atop 4} \right\} \approx 10^{59}$.

feasibly and thoroughly experiment with and reason about. Furthermore, many neural network ensembles introduced in research and used in practice are similarly homogenous, for instance, SnapShot Ensembles and Fast Geometric Ensembles (Huang et al., 2017a; Garipov et al., 2018). Additionally, our method provides a deterministic procedure of going between single network models and ensembles given a certain amount of parameters. Major sources of diversity in neural network ensembles are random weight initialization and stochastic training, both of which we incorporate in our framework.

**Depth-equivalent and width-equivalent ensembles.** Convolutional neural network architectures are determined by two design knobs – the depth and the width factor. Corresponding to these two design knobs, we create two classes of ensembles: depth-equivalent ensembles and width-equivalent ensembles. These are depicted in Figure 1: In depth-equivalent ensembles, the depth of the individual ensemble networks is the same as $S$ (i.e., $d$), and the width factor is set to the highest possible value (i.e., $w'$) without exceeding the parameter budget of $|S|$. In width-equivalent ensembles, on the other hand, the width factor is conserved across all ensemble networks (i.e., $w$), and the depth is modulated to the highest possible value (i.e., $d'$) without exceeding $|S|$:

$$w' : k \cdot |E_i^{(w',d)}| \leq |S^{(w,d)}| \leq k \cdot |E_i^{(w'+1,d)}| \qquad d' : k \cdot |E_i^{(w,d')}| \leq |S^{(w,d)}| \leq k \cdot |E_i^{(w,d'+1)}|$$

The above definition follows that neural networks in depth-equivalent ensembles have higher depth than those in width-equivalent ensembles. Width-equivalent ensembles contain wider neural networks than their depth-equivalent counterparts. In this way, we isolate and study the effect of depth and width on ensemble accuracy and resource requirement.

Overall, our design space spans three classes of convolutional neural network designs: (i) single network models, (ii) width-equivalent ensembles, and (iii) depth-equivalent ensembles. Every class contains several model designs instantiated by the four-tuple $\{w, d, |S|, C\}$. We next describe how we designed an exhaustive experimental framework to cover various configurations of these four-tuples.

## 3  FRAMEWORK: DATA, ARCHITECTURES, AND METRICS

**Datasets and architectures.** We include widely-used state-of-the-art network architectures and data sets in our study. These include DenseNets, and (Wide) ResNets architectures as well as SVHN, C10 and C100, and ImageNet datasets. Table A in the appendix summarizes these networks and training data sets as well as corresponding hyperparameters. We implement our experimental framework in PyTorch and used an Nvidia V100 GPU to run all experiments.

**Evaluation metrics.** We study all three design classes – single network, width- and depth-equivalent ensembles – across five metrics: (i) generalization accuracy, (ii) training time per epoch, (iii) time to accuracy, (iv) inference time, and (v) memory usage. When considered together, these metrics provide a holistic picture of the quality and practicality of models.

## 4  ENSEMBLES OUTPERFORM SINGLE NETWORK MODELS AFTER A LOW TO MODERATE PARAMETER THRESHOLD

We observe that both classes of ensembles – depth- and width-equivalent – outperform single network models after a resource threshold. We call this threshold the Ensemble Switchover Threshold (EST). Beyond the EST, ensemble models achieve 1 to 3 percent lower test error rates (across various architectures and data sets) compared with single networks having the same number of parameters.

The EST appears consistently across a wide range of data sets and architectures (Figure 2(a) through Figure 2(f)) as well as ensemble sizes (Figure 4(a) through Figure 4(c) and appendix Figure B(a) through Figure B(c)). In these figures, we use discrete heat maps to visualize which of the three design classes – single network models (single), depth-equivalent ensembles (deq), and width-equivalent ensembles (weq) – dominates in terms of generalization accuracy for a given resource budget. This resource budget takes the form of the number of parameters (on the $x$-axis) and epochs (on the $y$-axis). We also mark areas where both classes of ensembles outperform single network models. Figure 3 shows the test error rates achieved on various data sets for DenseNet models. We present this metric for the rest of the architectures in Figure A in the appendix.

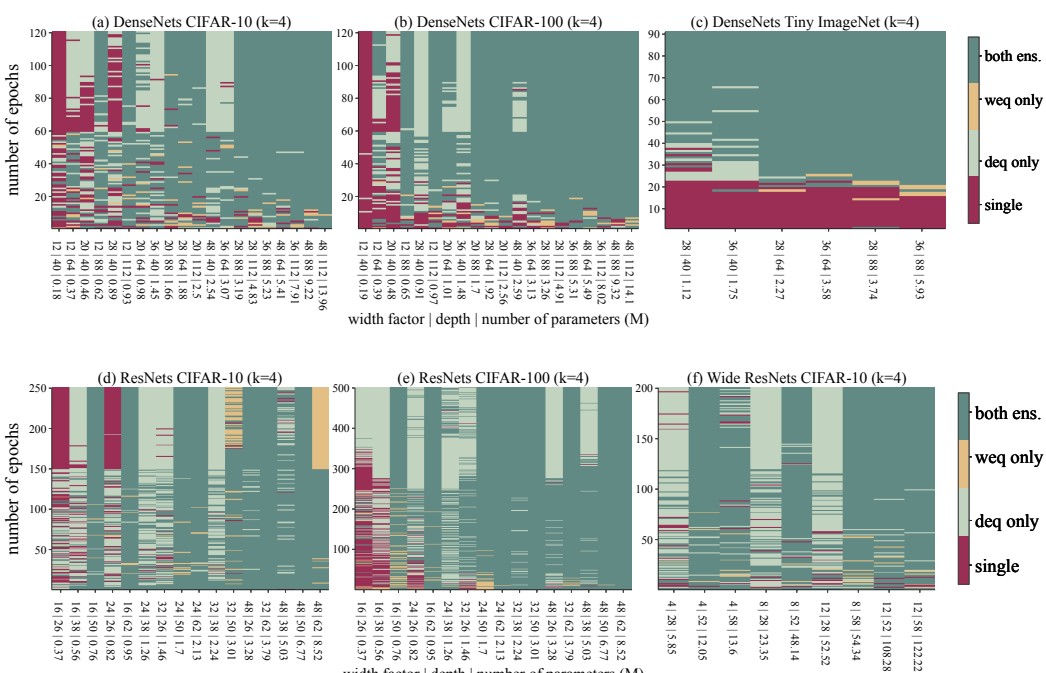

Figure 2: The Ensemble Switchover Threshold (EST) occurs consistently across various network architectures and data sets. Beyond this resource threshold, ensemble designs outperform single network models.

The occurrence of EST both expands and questions the general consensus on the relative effectiveness of ensemble versus single network models. First, even when allocated the same amount of resources, ensemble models still outperform single network models. This observation expands upon past empirical studies that only show how a $k$-network ensemble is more accurate than any of the single network models that it contains (Lee et al., 2015; Huang et al., 2017a). Second, the EST occurs in low- to moderate-resource settings. For instance, in all of our experiments, we observe the EST at the 1M to 1.5M parameter range[2] and after no later than half of the training epochs. This trend challenges the widespread notion that neural network ensembles are useful only when we have tons of resources at our disposal (Lee et al., 2015; Ju et al., 2017). Overall, our results indicate that *ensembles of convolutional models are preferable to single network models for a much wider range of use cases than previously understood.*

**On the superior generalization of ensembles under a parameter budget.** To interpret why ensembles outperform single network models under a parameter budget, we use the phenomenon of diminishing returns on increasing model sizes. In the past, this effect has been independently investigated by Eigen et al. (2013) and Dauphin and Bengio (2013) for a single network model. We hypothesize that as we increase the number of parameters, single network models exhibit more diminishing returns and plateau faster than ensemble networks. When the single network's generalization accuracy starts showing diminishing returns, the corresponding width-equivalent and depth-equivalent ensembles have smaller networks with $1/k$ as many parameters (assuming the parameters are spread equally along $k$ ensemble networks). These individual networks in the ensemble are affected less by the plateau because they have $1/k$ as many parameters as the single network model. Thus, utilizing these networks in an ensemble leads to better generalization accuracy overall because they do not hit the threshold of the diminishing returns while still being able to benefit from the known properties that ensembles provide: (i) They enrich the space of hypotheses that are considered by the base model class and (ii) By averaging over various models, ensembles reduce the variance of base models, smoothing out variations due to initialization and the learning process (Lee et al., 2015).

---

[2]Wide ResNets are an exception: This is because, compared to other convolutional architectures, Wide ResNets have an order of magnitude more parameters even for modest depths and width factors. For instance, networks presented in the Wide ResNet paper have anywhere between 8M to 37M parameters compared to the 1M to 10M range for DenseNets (Zagoruyko and Komodakis, 2016).

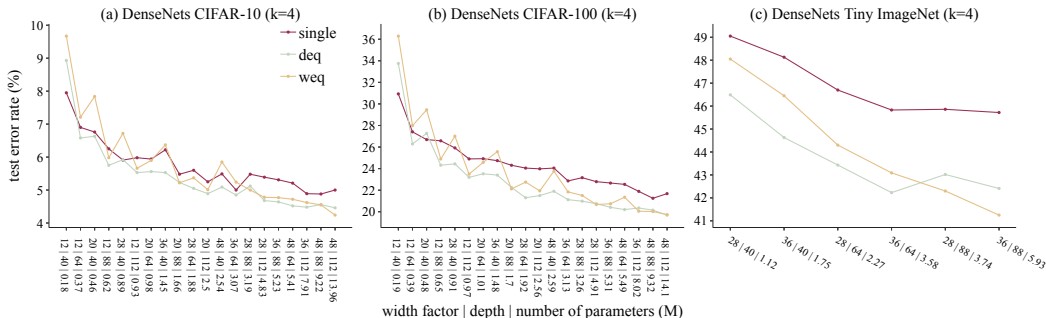

Figure 3: Ensembles arrive at lower test error rates than single network models after the EST has been reached.

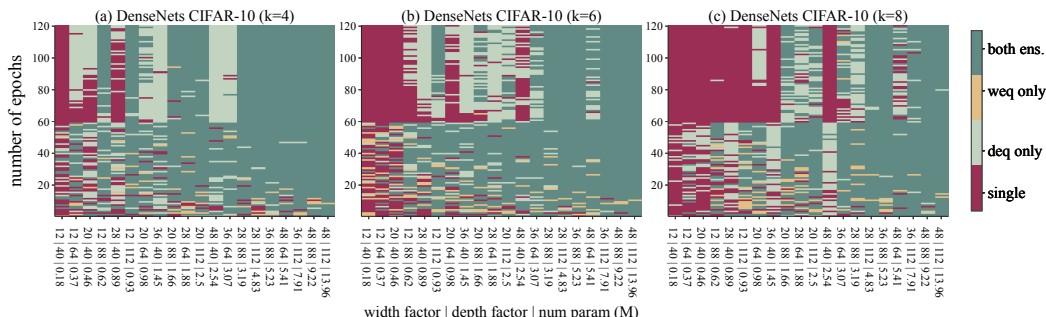

Figure 4: The Ensemble Switchover Threshold moves to the right as we increase the number of networks in the ensemble.

Next, we parse out how the data complexity and the composition of the ensemble networks affect the EST and, in turn, the ranking of the three design classes.

**Ensembles are even more effective for more complex data sets.** We observe that the EST shifts closer to the origin as the training data set's complexity increases. This can be seen in Figure 2(a) through 2(c) where we train DenseNets on progressively more complex data sets (CIFAR-10, CIFAR-100, and Tiny ImageNet). This observation indicates that ensemble models are preferable to single models when training on more complex data sets for an even wider range of available resources. This observation again expands the utility of ensembles. There is theoretical and empirical work establishing that ensembles do better for complex data (Bonab and Can, 2017; Huang et al., 2017a). We, however, establish this phenomenon in the resource-equivalent setting as opposed to past studies that do so for ensembles and single networks with drastically different numbers of parameters.

**Large ensembles are effective under a large parameter budget.** As we increase the number of networks $k$ within an ensemble without increasing the parameter budget, the overall accuracy of ensemble designs diminishes, pushing the EST to a higher resource limit. Figure 4 demonstrates this phenomenon for DenseNets and Figure B in the appendix shows it for ResNets. For instance, for DenseNet models, the EST moves from the $1.5M$ range for $k = 4$ to the $3M$ range for $k = 6$ and, then, to the $5M$ range for $k = 8$. This shift can be explained by looking at individual accuracy of ensemble networks. Figure C in the appendix shows test error rates of the ensemble as a whole as well as the average test error rates of individual ensemble networks corresponding to Figure 4. We observe that as we increase the number of networks $k$ (from $k = 4$ in Figure C(a) to $k = 8$ in Figure C(c)), the individual test error rates (shown as dotted lines) increases. This increase happens because individual networks' size goes down (as we keep a fixed parameter budget). This observation implies that larger-size ensembles are desirable over smaller sizes only when we have a sufficient parameter budget to assign to every single network in the model. As opposed to previous work, our experiments decouple the parameter budget from the number of networks in the model. We discover that just increasing the ensemble size without increasing the total number of parameters hurts accuracy.

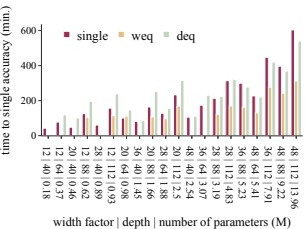

(a) DenseNets CIFAR-10 (k=4)

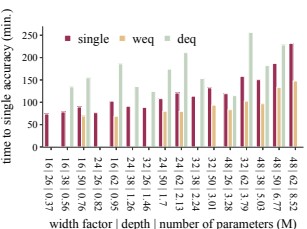

(b) ResNets CIFAR-10 (k=4)

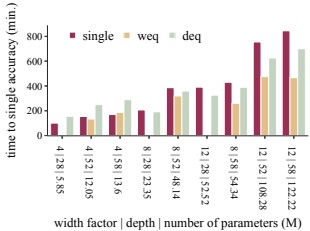

(c) Wide ResNets CIFAR-10 (k=4)

Figure 5: When ensemble designs can provide better accuracy, they can also do so faster than single network models (missing bars indicate that designs cannot reach single network model accuracy).

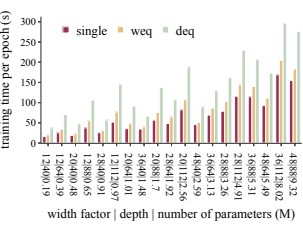

(a) DenseNets CIFAR-10 (k=4)

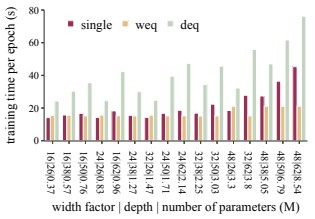

(b) ResNets CIFAR-10 (k=4)

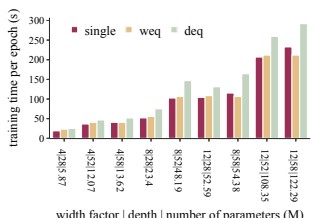

(c) Wide ResNets CIFAR-10 (k=4)

Figure 6: Depth-equivalent ensembles take longer to train per epoch as compared to single network models. Width-equivalent ensembles take comparable time.

**Depth-equivalent ensembles outperform width-equivalent ensembles.** We observe that depth-equivalent ensembles are overall more accurate than width-equivalent ensembles (as shown in Figure 3 and appendix Figure A). They also consistently demonstrate EST at a lower resource range. This can be explained by the fact that modern convolutional neural network architectures provide better accuracy with increasing depth (Eigen et al., 2013; Urban et al., 2016). Here, depth-equivalent ensembles have deeper ensemble networks with better individual accuracy. Thus, when used together in an ensemble, they also provide better ensemble accuracy. This way, when designing ensemble models for high accuracy, deeper networks are preferable to wider networks.

**EST vs. Memory Split Advantage.** For a limited part of the design space, recent work has observed the existence of a parameter limit beyond which depth-equivalent ensembles outperform single networks. This is termed as the Memory Split Advantage or the MSA (Kondratyuk et al., 2020). The EST, however, is not defined just with respect to the number of parameters but also the number of training epochs, inference cost, and memory usage as by only looking at these metrics in conjunction, we can get a complete picture of the relative effectiveness of ensembles vs. single networks. In this way, the EST subsumes the MSA, and we also verify the MSA across a significantly larger design space (e.g., we consider $3\times$ more data sets and twice as many architecture types) than has been done before.

## 5 ENSEMBLES TRAIN FASTER AND PROVIDE COMPARABLE INFERENCE TIME

First, we analyze the training time. Despite taking longer per epoch, both ensemble classes achieve the accuracy of single network models significantly faster for a considerable part of the design space (e.g., $1.2\times$ to $5\times$ faster across our experiments). This happens after the EST has been reached, i.e., *when ensemble designs can provide better accuracy, they can also do so faster than single network models*. This can be seen in Figure 5. Here, we plot the total training time needed for any of the three design classes to achieve the maximum accuracy of single network models under the same parameter budget. Figure 6 shows the corresponding training time per epoch.

**The combined depth determines per epoch training time.** We observe that both classes of ensembles, on average, take longer to train per epoch as compared to single network models as they train

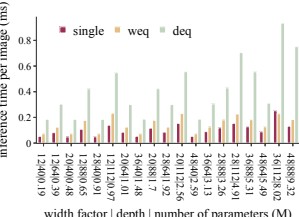 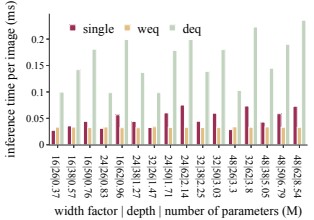 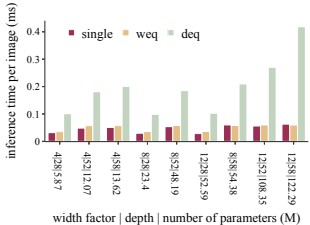

(a) DenseNets CIFAR-10 (k=4)    (b) ResNets CIFAR-10 (k=4)    (c) Wide ResNets CIFAR-10 (k=4)

Figure 7: Width-equivalent ensembles take comparable time to single network models for inference. Depth-equivalent ensembles take significantly longer.

$k$ networks instead of one. How much more time ensembles take per epoch depends heavily on the ensemble networks' design: This additional time is negligible for width-equivalent ensembles whereas, for depth-equivalent ensembles, it results in $2\times$ more expensive per epoch training. This trend can be explained by how the training time per epoch scales with respect to the width and depth of convolutional neural network models. This ultimately connects to how GPUs process data, which favors networks with few wider layers over those that have multiple thinner layers.

We break down the training time per epoch of all designs into two constituents: time spent per layer and number of layers. Figure E in the appendix shows this breakdown for various architectures. We observe that the total number of layers in a model (for ensembles, this is the sum of all networks' depth) majorly determines the training time per epoch. For the same parameter budget, depth-equivalent ensembles have proportionally more layers, whereas width-equivalent ensembles have proportionally more width. The average time per layer depends on the width and does not increase significantly as we move from depth-equivalent ensembles to width-equivalent ensembles. On the other hand, the total number of layers scales linearly with depth. For the same parameter budget, the total number of layers is significantly higher for depth-equivalent ensembles than the other two designs, resulting in higher per epoch training.

From a GPU perspective, wider and shallower networks are more efficient to execute than narrower and deeper networks for the same parameter budget. This can be attributed to the massive amount of data parallelism in modern GPUs. Increasing the network's width just increases the number of kernels within layers. This increase more efficiently utilizes GPU's massive capacity to perform the same operation on multiple data items. On the other hand, deepening a network introduces new layers (and operations) that require additional synchronization steps slowing down the overall execution.

**Networks in ensemble models converge faster than single network models for the same parameter budget.** The fact that ensemble designs can reach the same accuracy faster than single network models can be attributed to the fact that, for the same parameter budget, all networks in the ensemble model are smaller than the single network model. Smaller networks are known to converge faster albeit to lower accuracy than larger networks.

However, we observe that the distinct advantage ensemble designs provide over the single model is that when we use smaller networks in an ensemble, we get the best of both worlds. We converge faster at an individual network level, and ensembling makes up for the generalization accuracy.

Overall, these observations again question the conventional wisdom of ensembles being significantly slower to train than single network models. When we analyze the design space under a fixed parameter budget, we uncover that for a vast range of the design space: (i) width-equivalent ensembles introduce negligible overhead to per epoch training time as compared to single network models and (ii) both ensemble designs achieve and surpass accuracy of single network models in considerably less training time.

**Width-equivalent ensembles provide competitive inference time.** We provide the inference time per image in Figure 7 and observe a similar trend to training time per epoch. While depth-equivalent ensembles are significantly slower, *width-equivalent ensembles provide comparable inference speed to single network models.* Again, this questions conventional wisdom that expects ensembles to be substantially slower in inference.

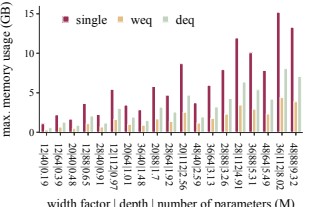 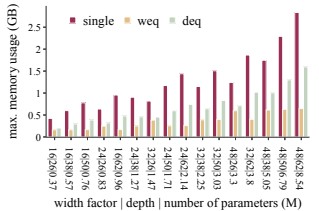 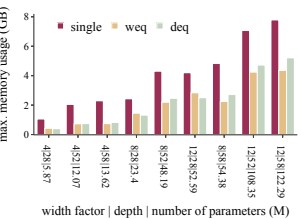

(a) DenseNets CIFAR-10 (k=4)   (b) ResNets CIFAR-10 (k=4)   (c) Wide ResNets CIFAR-10 (k=4)

Figure 8: Both classes of ensemble models are significantly more memory efficient.

## 6   ENSEMBLES ARE MEMORY-EFFICIENT

Regarding memory usage we observe that the trend favors both classes of ensemble designs over single network models. Figure 8 provides the amount of memory used as we train depth-equivalent ensembles, width-equivalent ensembles, and single network models. This is the minimum amount of memory that a GPU needs to train any of these designs for the batch sizes provided in Table A. This memory is majorly used to store model parameters and intermediate results (Jain et al., 2020).

The superior memory efficiency of ensemble models is because when we train a $k$-networks ensemble, at any point during the training process, we only need as much memory to train one of the $k$ networks (having $\frac{1}{k}$ as many parameters compared to the single network). This observation has two important implications: First, we can use larger batch sizes for the same GPU while training an ensemble of networks. This, for instance, is useful when training complex data sets such as ImageNet (Smith et al., 2018). Additionally, we can feasibly train the same number of parameters in an ensemble using lower-end GPUs with less memory.

**Additional results.** We show that the same resource-related trends hold for the rest of architectures and data sets from Table A. Figure D, Figure F, Figure G, and Figure H demonstrate these trends for metrics of time to accuracy, time per epoch, inference time, and memory usage respectively. We also provide results on a downsampled version of ImageNet-1K data set, CIFAR-100 with data augmentation, and SVHN data sets in Figure I, Figure J, and Figure L in the appendix respectively.

Table B through Table D provide the conversion between the width factor and depth of single network models and the two ensemble design classes.

## 7   THE COMPLETE PICTURE

The results presented question conventional wisdom concerning the design decision on whether to use an ensemble of convolutional networks or not. By creating a detailed framework that (a) allows to fix resources and (b) spans a large design space, we show that for a considerable part of the design space, given an amount of resources, ensembles (i) achieve better accuracy than single network models, (ii) train faster, (iii) provide comparable inference, and (iv) need much less memory. Future work includes the addition of fast ensemble training approaches (such as SnapShot, TreeNets, and MotherNets (Lee et al., 2015; Huang et al., 2017a; Wasay et al., 2020)) and parallel training, both of which can move EST further in favor of ensembles.

**Limitations and future directions.** The analysis framework as presented in this paper covers extensively homogeneous ensembles. There are several opportunities for parts of the design space not covered by this work where it would be extremely useful to verify the results further. In particular, this includes (1) heterogeneous ensembles (e.g., various ratios of width-equivalent and depth-equivalent networks), (2) arbitrary network architectures (including using network architecture search approaches), and (3) application domains other than image classification such as object detection, machine translation, and deep generative models.

**Acknowledgements.** Harvard DASlab member Pablo R. Ruiz, Haochen Yang, Longshen Ou, and Ziyi Guo helped with parts of the implementation and the demo. This work is partly funded by the USA Department of Energy Project DE-SC0020200.

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

# APPENDIX

Table A: For all networks, we use training hyperparameters listed in their respective papers (lr: learning rate, wd: weight decay).

| Architecture | Data sets | Epochs | Lr schedule | batch size | Wd |
|---|---|---|---|---|---|
| DenseNet | SVHN | 100 | 0.1, 0.01(30), 0.001(60) | 128 | 0.001 |
| DenseNet | C10 and C100 | 120 | 0.1, 0.01(60), 0.001(90) | 128 | 0.001 |
| DenseNet | Tiny ImageNet | 90 | 0.1, 0.01(30), 0.001(60) | 64 | 0.001 |
| DenseNet | ImageNet32-1K | 90 | 0.1, 0.01(30), 0.001(60) | 64 | 0.001 |
| ResNet | C10 | 250 | 0.1, 0.01(100), 0.001(200) | 128 | 0.0001 |
| ResNet | C100 | 500 | 0.1, 0.01(250), 0.001(375) | 128 | 0.0001 |
| Wide ResNet | C10 and C100 | 200 | 0.1, 0.01(100), 0.001(200) | 128 | 0.0001 |

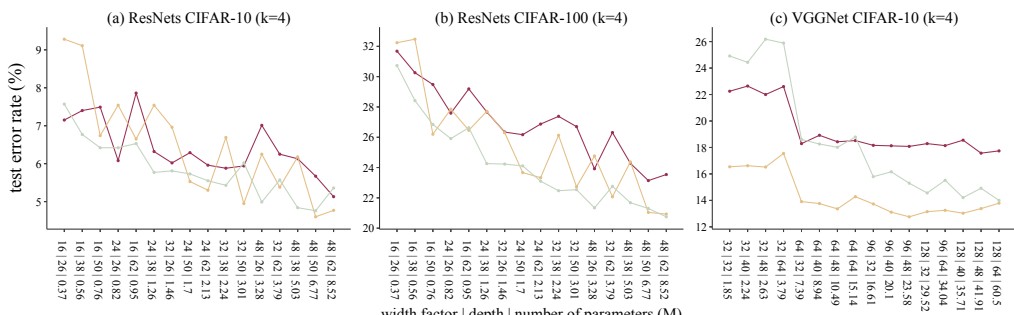

Figure A: Ensembles arrive at lower test error rates than single network models after the EST has been reached.

## A  ON THE INSTABILITY OF WIDTH-EQUIVALENT ENSEMBLES

The performance of width-equivalent ensembles (weq ensembles) exhibits unstable behavior. In particular, it has local spikes when it comes to the test error rates. This is particularly pronounced for the ResNet ensembles (Figure A(a) and Figure A(b)).

Our interpretation of this phenomenon is that these local spikes have to do with the relative depth of networks in the width-equivalent ensemble designs. When comparing designs that are close together in the parameter range, we observe that the designs with more depth (of ensemble networks) generally outperform those with less depth even if the latter have more parameters. The depth of networks in the weq ensemble plays a more dominant role than the total number of parameters they have. As an example of this, consider Figure A(a) and A(b) in the revised version of the paper (ResNet CIFAR-10 (k=4) and ResNet CIFAR-100 (k=4). Weq exhibit three spikes at 2.24M, 3.28M, and 5.03M parameters. All three of these points are flanked on both sides by designs that have similar number of parameters but more depth.

This observation is consistent with past observations that depth is more influential in determining the accuracy of networks (Eigen et al., 2013).

## B  ADDITIONAL EXPERIMENTS IN UNDERFITTING SCENARIOS

We now provide experiments on two additional data sets from underfitting scenarios: (i) ImageNet-1K-32 a downsampled version of ImageNet-1k with 1.2 million images and 1000 labels (every image downsampled to be of size 32 × 32), and (ii) CIFAR-100 with aggressive rand-augment (Cubuk et al., 2020).

The result of these experiment can be seen in Figure I and Figure J respectively. Overall, our observations hold for these data sets as well. After a certain parameter budget (defined in terms of number of parameters and training epochs), ensembles of neural networks consistently outperform the single network and also reach that accuracy faster when compared to the single model.

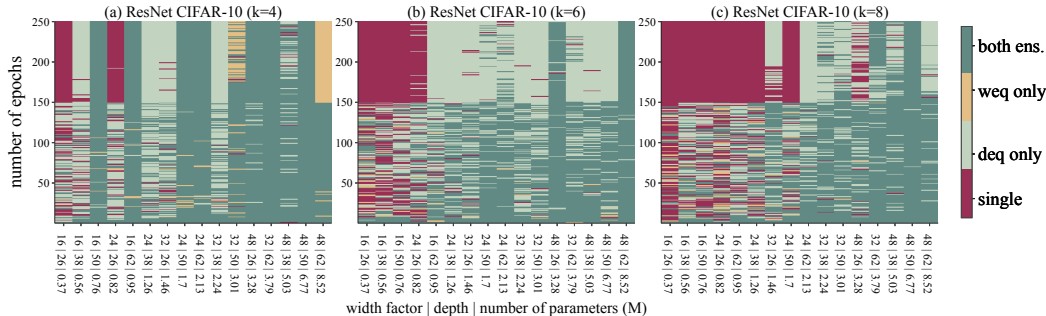

Figure B: The Ensemble Switchover Threshold moves to the right as we increase the number of networks in the ensemble. Here, we demonstrate this phenomenon for ResNet models.

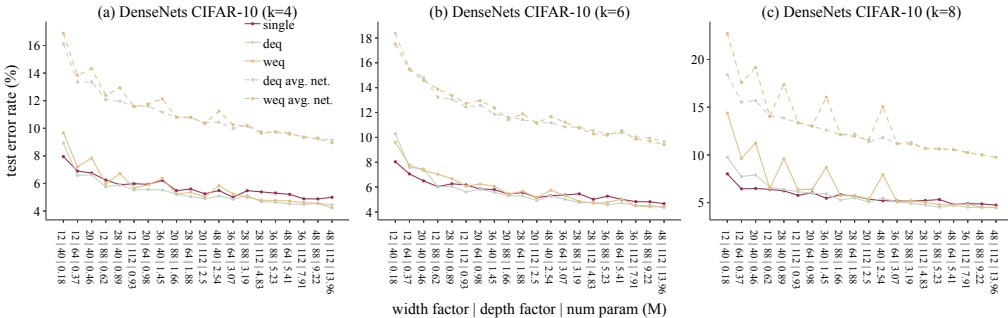

Figure C: As we increase the size of ensembles, accuracy of individual networks in the ensemble decreases. This results in an overall reduction in ensemble accuracy shifting the EST to the high-resource space.

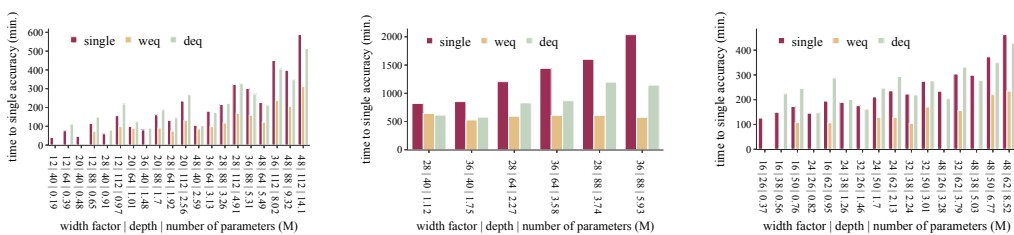

(a) DenseNets CIFAR-100 (k=4)    (b) DenseNets Tiny ImageNet (k=4)    (c) ResNets CIFAR-100 (k=4)

Figure D: When ensemble designs can provide better accuracy, they can also do so faster than single network models (missing bars indicate that designs cannot reach single network model accuracy).

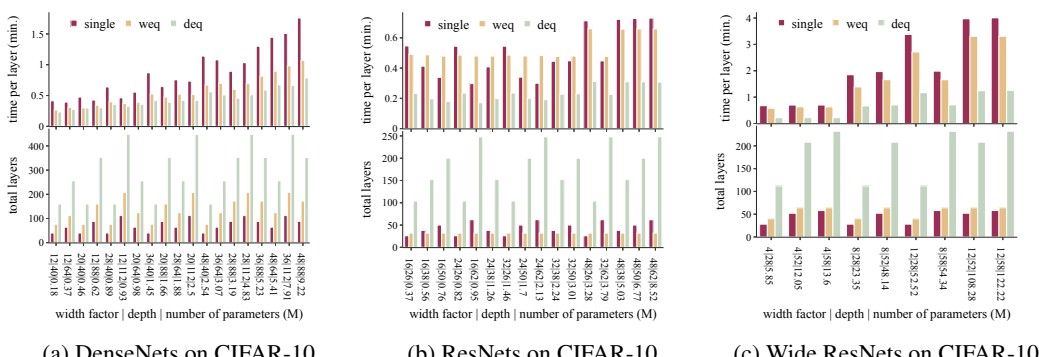

(a) DenseNets on CIFAR-10        (b) ResNets on CIFAR-10        (c) Wide ResNets on CIFAR-10

Figure E: We break down per epoch training time into: (i) time spent per layer and (ii) total number of layers. We observe that the total number of layers in the model more significantly determines the per epoch training time as compared to the width.

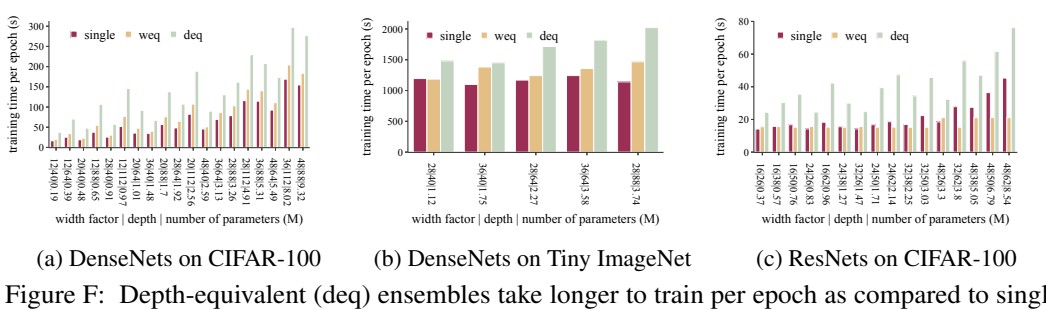

(a) DenseNets on CIFAR-100        (b) DenseNets on Tiny ImageNet        (c) ResNets on CIFAR-100

Figure F: Depth-equivalent (deq) ensembles take longer to train per epoch as compared to single network models. Width-equivalent ensembles (weq) take comparable time.

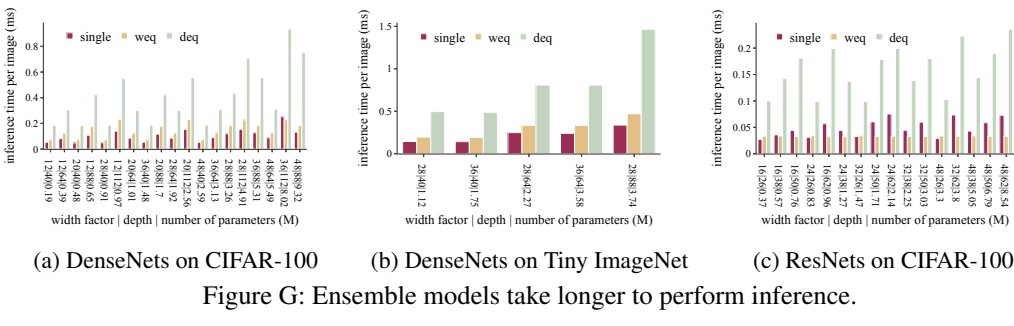

(a) DenseNets on CIFAR-100        (b) DenseNets on Tiny ImageNet        (c) ResNets on CIFAR-100

Figure G: Ensemble models take longer to perform inference.

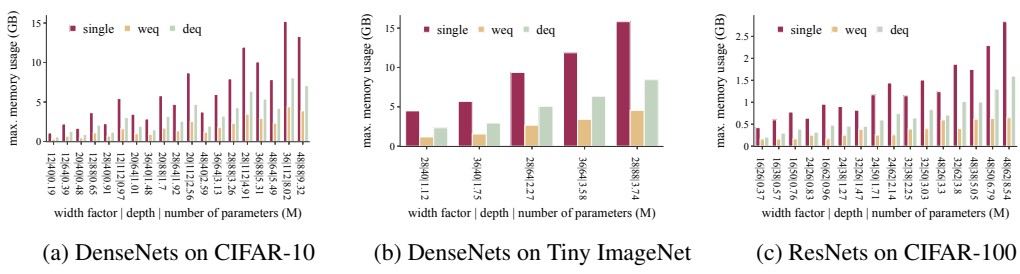

(a) DenseNets on CIFAR-10        (b) DenseNets on Tiny ImageNet        (c) ResNets on CIFAR-100

Figure H: Ensemble models are significantly more memory efficient.

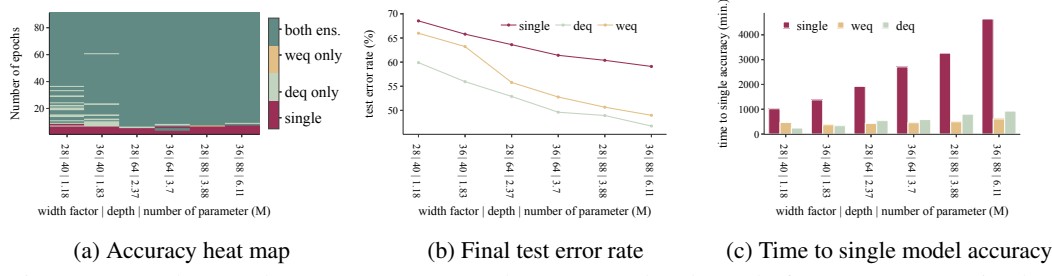

(a) Accuracy heat map     (b) Final test error rate     (c) Time to single model accuracy

Figure I: We observe the same accuracy and resource related trends for DenseNets trained on ImageNet32, a downsampled version of the ImageNet-1k data set (Chrabaszcz et al., 2017).

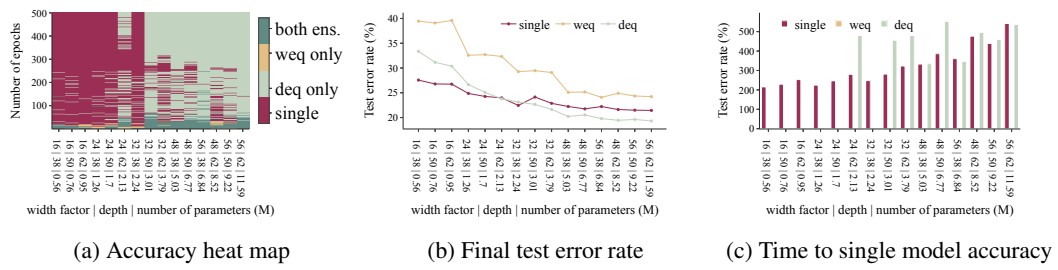

(a) Accuracy heat map     (b) Final test error rate     (c) Time to single model accuracy

Figure J: We observe the same accuracy and resource related trends for ResNets trained on CIFAR-100 with agressive data augmentation to mimic underfitting scenario. In particular, we use RandAugment with $N = 2$ and $M = 14$ (Cubuk et al., 2020).

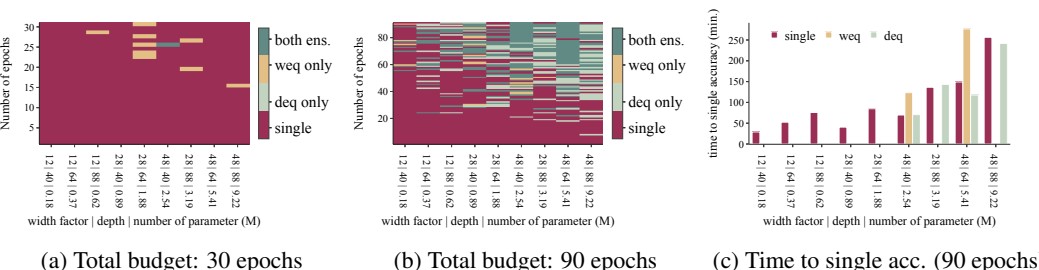

(a) Total budget: 30 epochs     (b) Total budget: 90 epochs     (c) Time to single acc. (90 epochs)

Figure K: We repeat our experiments with different total training budgets and set up the learning rate schedule proportionally for every budget. Here we show results for training DenseNets on the CIFAR-10 data set. We observe the same trend as the total budget increase, ensembles provide better accuracy and can train faster.

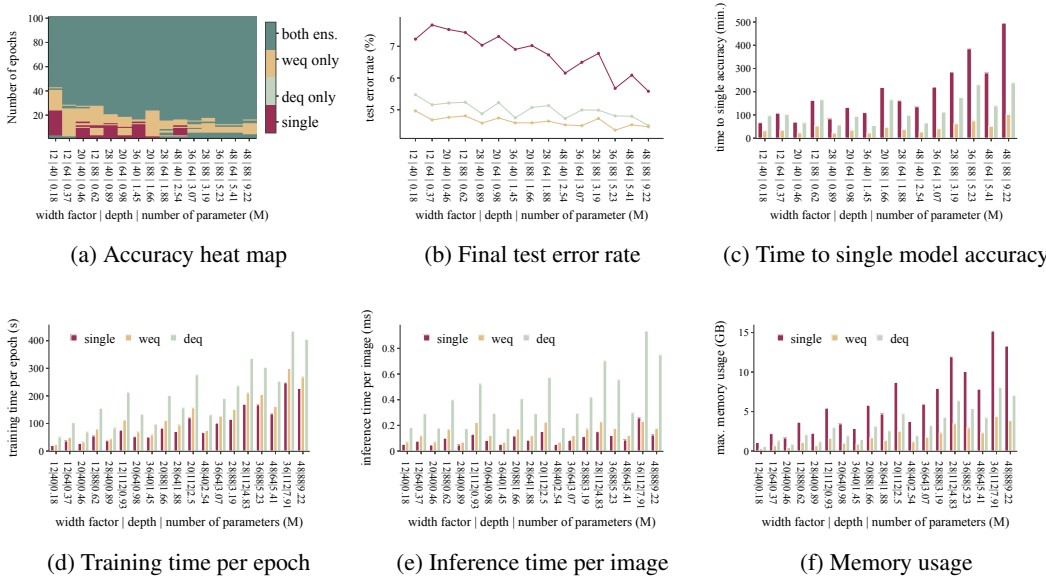

(a) Accuracy heat map     (b) Final test error rate     (c) Time to single model accuracy

(d) Training time per epoch     (e) Inference time per image     (f) Memory usage

Figure L: We observe similar accuracy and resource related trends for DenseNets trained on the SVHN data set.

Table B: DenseNet

| single depth | single width factor | Weq depth | Deq width factor |
|---|---|---|---|
| 40 | 12 | 19 | 5 |
| 40 | 20 | 19 | 9 |
| 40 | 28 | 19 | 13 |
| 40 | 36 | 19 | 17 |
| 40 | 48 | 19 | 23 |
| 64 | 12 | 28 | 6 |
| 64 | 20 | 31 | 10 |
| 64 | 28 | 31 | 14 |
| 64 | 36 | 31 | 18 |
| 64 | 48 | 31 | 24 |
| 88 | 12 | 40 | 6 |
| 88 | 20 | 40 | 10 |
| 88 | 28 | 43 | 14 |
| 88 | 36 | 43 | 18 |
| 88 | 48 | 43 | 24 |
| 112 | 12 | 52 | 6 |
| 112 | 20 | 52 | 10 |
| 112 | 28 | 52 | 14 |
| 112 | 36 | 52 | 18 |

Table C: ResNet

| single depth | single width factor | Weq depth | Deq width factor |
|---|---|---|---|
| 26 | 16 | 8 | 8 |
| 26 | 24 | 8 | 12 |
| 26 | 32 | 8 | 16 |
| 26 | 48 | 8 | 24 |
| 38 | 16 | 8 | 8 |
| 38 | 24 | 8 | 12 |
| 38 | 32 | 8 | 16 |
| 38 | 48 | 8 | 24 |
| 50 | 16 | 14 | 8 |
| 50 | 24 | 14 | 12 |
| 50 | 32 | 14 | 16 |
| 50 | 48 | 14 | 24 |
| 62 | 16 | 14 | 8 |
| 62 | 24 | 14 | 12 |
| 62 | 32 | 14 | 16 |
| 62 | 48 | 14 | 24 |

Table D: WideResNet

| single depth | single width factor | Weq depth | Deq width factor |
|---|---|---|---|
| 28 | 4 | 10 | 1 |
| 28 | 8 | 10 | 3 |
| 28 | 12 | 10 | 5 |
| 52 | 4 | 16 | 1 |
| 52 | 8 | 16 | 3 |
| 52 | 12 | 16 | 5 |
| 58 | 4 | 16 | 1 |
| 58 | 8 | 16 | 3 |
| 58 | 12 | 16 | 5 |

