# OpenReview forum: "More or Less: When and How to Build Convolutional Neural Network Ensembles"
_ICLR.cc/2021/Conference — ICLR 2021 Poster_

### Official Review · AnonReviewer3 · 2020-10-27
**The paper provides a comprehensive evaluation of building ensemble networks compared to single network while keep the number of paramters equal. The experiments are conducted on four datasets. The comparison is conducted for test error, training time per epoch, time to optimization, inference time and memory usage.**

**Rating:** 7
**Confidence:** 4

**Review:**

The paper provides a comprehensive evaluation of building ensemble networks compared to a single network while keeping the number of parameters equal. The experiments are conducted on four datasets. The comparison is conducted for test error, training time per epoch, time to optimization, inference time, and memory usage.

The paper is well written and it is a pleasure to read it. The experiments are thorough and comparisons are thoughtful. One minor comment on readability is that a lot of results are pointing to the appendix. However, I do understand that this is due to the limited space. The authors may try to move similar results to the appendix and bring the other important results back in the paper. For example, it might be okay to limit figure 2 to three diverse plots instead of all 6 plots.

Minor comments/questions:

- Width equivalent ensemble shows quite an unstable result (Figure A). Do authors have any intuition about it?
- I would like to see more discussion on the generalization capability of ensemble versus individual networks. Authors have mentioned it as one of their evaluation metrics but very little light is shed on this.

---

> ### Author Response · Authors · 2020-11-21
> **Instability of Weq addressed | Intuition on generalization capabilities added| Presentation revised**
>
> We appreciate the excellent feedback provided by the reviewer. We address all of the reviewer's comments below and we have revised the paper accordingly.
>
> **Suggestions to improve readability**
>
> We thank the reviewer for excellent suggestions to improve the presentation of the paper. In the revised version, we have moved Fig. A (a through c) to the main paper. This brings an additional important result to the main paper from the appendix and allows us to drop many of the appendix references.
>
> In addition, since we have an extra page available for the rebuttal period, we kept all six plots in Figure 2 to highlight the fact that EST occurs across a variety of data sets and architectures. In case we run short on space if more changes are needed, we will move three plots from Fig. 2 to the appendix as suggested by the reviewer.
>
> **On the instability of width-equivalent ensembles**
>
> This is a great observation. Thanks for bringing this up.
>
> The performance of width equivalent ensembles (weq ensembles) is indeed unstable. In particular, it has local spikes when it comes to the test error rate. This is particularly pronounced for the ResNet ensembles (Figure A(a) and Figure A(b) in the revised version of the paper).
>
> Our interpretation of this phenomenon is that these local spikes have to do with the relative depth of networks in the weq ensemble designs. When comparing designs that are close together in the parameter range, we observe that the designs with more depth (of ensemble networks) generally outperform those with less depth even if the latter have more parameters. The depth of networks in the weq ensemble plays a more dominant role than the total number of parameters they have. As an example of this, consider Figure A(a) and A(b) in the revised version of the paper (ResNet CIFAR-10 (k=4) and ResNet CIFAR-100 (k=4). Weq exhibit  three spikes at 2.24M, 3.28M, and 5.03M parameters. All three of these points are flanked on both sides by designs that have similar number of parameters but more depth.
>
> In this way, the test error spikes observed for width equivalent ensembles follow the relative depth of the respective networks. This observation is consistent with past observations that depth is more influential in determining the accuracy of networks [1].
>
> We now include a paragraph in Appendix Sec. A to explain this observation. Thanks again for bringing our attention to this phenomenon.
>
> [1] Understanding Deep Architectures using a Recursive Convolutional Network (https://arxiv.org/pdf/1312.1847.pdf)
>
> **On generalization capabilities of ensemble architectures**
>
> In the original submission we included as a measure of generalization capability the final test error rate of all experiments. This was in the appendix. In the revised version of the paper, we have moved these results in the main paper in Figure 3 and there are additional results in Figure A.
>
> In addition we now include a discussion that explains these results in terms of why we see ensembles improving in generalization.  This discussion can be seen in Sec. 4 under “On superior generalization capability of ensembles under a parameter budget”.
>
> Our interpretation for why ensembles outperform single networks on a parameter budget is that as we keep on increasing the size of a single convolutional neural network we run into diminishing returns and the generalization capability of a single network hits a plateau. In past analysis of single networks, this effect has been independently investigated by [1] and [2]. With ensambles, when the generalization accuracy of the single network starts showing diminishing returns, the corresponding width-equivalent and depth-equivalent ensembles have smaller networks with 1/k as many parameters (assuming the parameters are spread equally along k ensemble networks). These individual networks in the ensemble are affected less by the plateau at this point because they have 1/k the parameters. Thus, utilizing these networks in an ensemble leads to better generalization accuracy overall because they do not hit the diminishing returns threshold, while still being able to utilize the known properties that ensembles provide: (i) They enrich the space of hypotheses that are considered by the base model class (ii) By averaging over various models, ensembles reduce the variance of base models, smoothing out variations due to initialization and the learning process [3].
>
> [1] Big neural networks waste capacity. (https://arxiv.org/pdf/1301.3583.pdf)
>
> [2] Deep Double Descent: Where Bigger Models and More Data Hurts (https://arxiv.org/pdf/1912.02292.pdf)
>
> [3] Why M Heads are Better Than One: Training a Diverse Ensemble of Deep Networks (https://arxiv.org/pdf/1511.06314.pdf)
>
> **.: Changes to the paper :.**
>
> (1) Fig. A (a through c) is now Fig. 3 (a through c)
>
> (2) Added Sec. A on “Instability of Width Equivalent Ensembles”
>
> (3) Add a paragraph in Sec. 4 “On superior generalization of ensembles under a parameter budget”

---

### Official Review · AnonReviewer4 · 2020-10-28
**Final review from R4**

**Rating:** 5
**Confidence:** 4

**Review:**

This paper establish a robust and holistic framework to compare scaling up an ensemble with scaling up a single networks, where test accuracy, number of paramaters, inference time, memory consumption and training time to converge are considered.  To reduce the intractably large design space of scaling up an ensemble, the author mainly investigate two types of ensembles: depth-equivalent and width-equivalent ensembles.  Through extensive experiments on SVHN, CIFAR-10, CIFAR-100, and Tiny ImageNet with VGGNets, ResNets, DenseNets and WideResNets, the authors discovered an surprising and consistently emerging phenomenon named The Ensemble Switchover Threshold: When the amount of resources (measured by number of parameters, training cost) is beyond this threshold, ensembles methods provide better performance and computation trade-off.

Overall, I recommend this paper to be accepted because:

1. This is the first paper to  conduct an extensive and robust comparison between scaling up a single model and scaling up an ensemble.
2. The EST phenomenon uncovered by this work is supervising and country to common knowledge.

My major concerns:
1. Though the author conducted extensive results, but most of the experiments are conducted in the scenario where overfitting is the major problem affecting the test accuracy rather than under-fitting due to the authors' limited computing resources(One V100 as mentioned).  I am interested if the EST still holds in under-fitting scenarios.  On possible way to investigate this phenomenon in the under-fitting without huge training cost is to mimic the under-fitting scenarios by carrying out aggressive data augmentations during training, like RandAug[1].
2. On the details of measure the training time needed for different models to reach a fixed performance in your figure 6.  As you are using the step-wise constant learning rate schedule (Tabel A in appendix), which easily leads to a flattened loss curve during training. To measure the performance of one method with a set of training budgets,  you should adjust the learning rate schedule accordingly and measure the performance separately in different runs.  It maybe be wrong if you are measuring the performance of different checkpoints generated in one run.

[1] Cubuk, Ekin D., et al. "Randaugment: Practical automated data augmentation with a reduced search space." Proceedings of the IEEE/CVF Conference on Computer Vision and Pattern Recognition Workshops. 2020.

Overall, it is an interesting paper.

==================Post-discussion Update===========================

Thanks to the authors for addressing my concerns.

However, after viewing the other responses (especially the comments from Ekaterina Lobacheva) as well as the author's explanation, I think this submission missed out some quite important references. In addition, its contribution over previous works appears to be marginal after viewing these references. Therefore, I would like to lower my rating from 6 to 5.

---

> ### Author Response · Authors · 2020-11-21
> **Two new experiments on underfitting scenarios added | Discussion on learning rate schedule**
>
> We thank the reviewer for their excellent and thorough feedback.
>
> **Additional Experiments on underfitting scenarios**
>
> This is a fair observation. We also thank the reviewer for their helpful suggestion on using random augmentation to feasibly mimic an underfitting scenario.
>
> We now provide experiments on two additional data sets from underfitting scenarios:
>
> (i) CIFAR-100 with aggressive rand-augment as suggested by the reviewer, and
>
> (ii) ImageNet-1K-32,  a downsampled version of ImageNet-1k with 1.2 million images and 1000 labels [1].
>
> The results of these experiments can be seen in Figure I and Figure J in the appendix of the revised version of the paper.  For ImageNet-1K-32, we include 4 different parameter budgets and for CIFAR-100 with aggressive data augmentation  we report 12 parameter budgets. Overall, we observe the same trends as with the original results. After a certain parameter budget (defined in terms of number of parameters and training epochs), ensembles of neural networks consistently outperform the single network and also reach that accuracy faster when compared to the single model.
>
> By the end of the review period, we expect to have results for two more parameter budgets for  ImageNet-1K-32 and 3 more for CIFAR-100 with aggressive data augmentation. We will update the paper as these results become available.
>
> [1] A Downsampled Variant of ImageNet as an Alternative to CIFAR datasets (https://arxiv.org/abs/1707.08819)
>
> **On learning rate schedule**
>
> This is an excellent observation. We should have made that clear as part of our setup in the first place.
>
> We agree with the reviewer that to fully capture the performance profile of models for a given set of training budgets, while using a step-wise learning rate schedule, we would need to start the experiment from scratch every time and schedule the step-wise learning rate for every training budget.
>
> We now describe why we had to do something different. Please note that the design space we cover across architectures (tunings), data sets, and metrics is massive. Our experiments take a full GPU year to run. If we were to reset the learning rate and rerun for every epoch across our experiments then it would take us 60 times more time to complete our analysis for DenseNets (for the maximum of 120 epochs as in our current setup).
>
> Since this would make the analysis infeasible, we made the decision to train to 120 epochs and observe results as they evolve with a learning rate schedule set for the target 120 epochs.
>
> For this methodology we draw inspiration from the time to accuracy metric used in DawnBench, a public benchmark to compare the end-to-end performance of deep learning models [1]. This metric compares in a black box way the time taken by any two models to arrive at a specified accuracy. This metric answers the question that if we use ensemble models, how much time will it take them to reach the final accuracy of a single neural network model having the same number of parameters.
>
> We now include an explanation of this decision in the experimental setup under “Evaluation metrics” in Section 3.
>
> In addition, we plan to run an additional experiment in the next few days as per the reviewers suggestion. We will do that by “sampling” from the massive set of possible experiments so we can get an indication of behavior within a reasonable timeframe. We will pick a small set of training budgets (in terms of target epochs) and for every training budget, we will set up the learning rate schedule independently and train separately from scratch.
>
> We thank the reviewer again for this remark to improve the depth of our study.
>
> [1] DAWNBench: An End-to-End Deep Learning Benchmark and Competition (https://dawn.cs.stanford.edu/benchmark/papers/nips17-dawnbench.pdf)
>
> **.: Changes to the paper :.**
>
> (1) Added Figure I and Figure J in the appendix: experiments added on underfitting scenarios: ImageNet32 and CIFAR-100 with aggressive data augmentation.
>
> (2) Added Section B in the appendix on “Additional Experiments in Underfitting Scenarios”
>
> (3) Explained the time to accuracy metric under “Evaluation Metric” scenario in Section 3

---

> > ### Author Response · Authors · 2020-11-25
> > **New parameter budgets on ImageNet32 and CIFAR-100 (RandAugment) added | Resetting learning rate experiments added**
> >
> > We have now added results from two more parameter budgets (3.7M and 6.11M) for DenseNets trained on ImageNet32  and three more parameter budgets (6.84M, 9.22M, and 11.59M) for ResNets trained on CIFAR-100 with aggressive randaugment. This can be seen in Figure I and Figure J  in the appendix respectively. The overall trends in accuracy as well as training time hold for these two additional data sets from overfitting scenarios.
> >
> > We have also added new experiments, where we pick a sample of training budgets (in terms of target epochs) and for every training budget, we set up the learning rate schedule independently and train separately from scratch. We experiment with a training budget of 30 and 90 training epochs on top of the original setup of 120 epochs. This is using DenseNets on CIFAR-10. Figure K in the latest version of the paper shows the results for these experiments. We observe the same trend as the total budget increases, ensembles provide better accuracy and can train faster.
> >
> > We would like to thank the reviewer again for their helpful comments.

---

### Official Review · AnonReviewer2 · 2020-10-28
**Demonstration of when to use ensemble methods of CNNs**

**Rating:** 8
**Confidence:** 3

**Review:**

Summary: This paper addresses when to use a single network model vs an ensemble of convolutional neural network models based on resource budgets. The authors challenge the notion that ensemble methods should only be used when resources are a non-issue. The authors compare single networks to width-equivalent and depth-equivalent ensemble methods for SVHN, cifar10, cifar100 and tiny imagenet across multiple network architectures and describe the 'Ensemble Switchover Threshold (EST)', the amount of resources beyond which ensembles provide better generalization accuracy than single models.

Strengths:
The authors robustly test this threshold where an ensemble method outperforms single network models.

The authors consider more than just accuracy but also at inference cost and memory usage which are important parameters for deployable code.

The provided demo is an impressive and straight-forward visualization tool for understanding when to use which type of model.

The authors explore how performance changes across number of models in the appendix -- a question I thought of while reading the paper and did not expect to get answered.

Weaknesses:
There are some inevitable limitations to this type of study that make the EST hard to interpret. The authors do not include heterogenous ensemble methods or other hyper-parameterization that might vary between the three setups: single model, width and depth-equivalent.

The title and majority of the abstract brag a great scope than the paper considers. It should be made clear in the title that only CNNs are considered for this study. The work done within this scope is thorough and impressive, and stands alone in its value.

Questions:
The authors assumption that the number of parameters is directly proportional to the resources used seems reasonable but I did wonder if there was a citation or related work to back this point up?

---

> ### Author Response · Authors · 2020-11-21
> **Added limitations and future work | Added citations | Revised presentation**
>
> We would like to thank the reviewer for their constructive feedback. We address all the points raised by the reviewer below.
>
> **Heterogeneous ensembles**
>
> This is definitely a valid point regarding the scope of our study. We would like to note that the possible space of all ensembles (homogeneous and heterogeneous) given a single neural network is extremely large (as we discuss in Section 2 under “The single network versus ensemble design space”). Our approach to understanding the space and to start extracting patterns is to perform a detailed study across numerous metrics that give a holistic view. As such, experiments take a long time to run and so we had to select an interesting part of the space where results would still be meaningful and would help push the understanding of our community on the topic.
>
> We have now included a new paragraph towards the end of the paper where we highlight limitations and next steps. We specifically bring up: 1) heterogeneous ensembles, 2) ensembles with diverse network architectures, as well as 3) additional application spaces.
>
> Thanks much for the remark.
>
> **Citations on the number of parameters are proportional to resource utilization**
>
> Thanks for bringing our attention to this. There is indeed recent related work that independently supports the observation that the number of parameters is proportional to resource utilization for a given neural network architecture. In particular, this has been shown for inference and training time [1] as well as memory usage [2]. We now include references to these papers in the revised version of the paper (Section 2 under “Establishing grounds for fair ranking”). Also, our experimental results verify this observation across all resource metrics: As we increase the number of parameters, the training time per epoch, the inference time, and the memory usage show an increasing trend. This can be seen in Figures 6, 7, and 8 in the revised version of the paper.
>
> [1] MotherNets: Rapid Deep Ensemble Learning (https://proceedings.mlsys.org/paper/2020/file/3ef815416f775098fe977004015c6193-Paper.pdf)
>
> [2] Checkmate: Breaking the Memory Wall with Optimal Tensor Rematerialization (https://proceedings.mlsys.org/paper/2020/file/084b6fbb10729ed4da8c3d3f5a3ae7c9-Paper.pdf)
>
> **Title and abstract should reflect the focus on convolutional architectures**
>
> This is a perfectly fair point. We have changed the title and the abstract to reflect that we focus on convolutional architectures.
>
> **.: Changes to the paper :.**
>
> (1) Section 7: Added a paragraph at the end of addressing limitations and future work.
>
> (2) Section 2 under “Establishing grounds for fair ranking”: Cited papers that elucidate the relationship between number of parameters and resource metrics
>
> (3) Title: Revised to “More or Less: When and How to Build Convolutional Neural Network Ensembles”
>
> (4) Abstract: Revised to reflect the focus on convolutional architectures.

---

### Official Review · AnonReviewer1 · 2020-10-28
**Well-designed exploration and evaluation of ensembles of smaller neural nets, showing they usually outperform single comparable networks.**

**Rating:** 8
**Confidence:** 4

**Review:**

Summary
--------------
The paper evaluates an under-explored space of neural models: ensemble of smaller networks (shallower or narrower). Extensive experiments show that when growing the total capacity (number of parameters) beyond a threshold, these ensembles get better performance than a single network, and train faster.

Pros
-------
- Sound methodology to quantify the "conventional wisdom" around ensembles of networks, and explore a larger design space
- Good experimental design and exploration, especially with limited resources (1 GPU-year)
- Really interesting main result, how ensembles typically get better at exploiting additional capacity than single networks, once the capacity is large enough

Cons
--------
- Code is not released
- Only small to medium datasets are used (nothing like the scale of ImageNet-21k for instance), so some of the observations may not hold for larger datasets or models (although the main conclusions are likely to)

Recommendation
---------------------------
I recommend **acceptance** of this paper, as it provides new insight on when to use ensembles of smaller models, and justifies it by exhaustive experiments. It provides a sound basis for exploring a wider space of neural-based systems, and understanding better the strong points of ensembles.

Arguments
----------------
- A better evaluation of ensembles or neural nets, as well as their trade-offs, is quite *significant* for the machine learning community, as it could change how we think of structuring systems.
- Thinking of exploiting ensembles of smaller models, and not only for scaling up beyond how one model can reasonably be, is novel and *original* to my knowledge, and goes against the usual principle of "jointly training everything" that is pervasive in the deep learning community. The results are somewhat unexpected, and clearly show the advantage of this approach.
- The *quality* of the methodology is really high, as it enabled exploration of a wide space in a reasonable and straightforward way (with architecturally-homogenous ensembles), when an exhaustive enumeration would be impossible. The metrics evaluated strongly support the conclusion.
- The design, experiments, and results are very clearly explained, are exposed in a straightforward way and easy to follow.

Questions
---------------
The main question I had reading the article was "how useful is that additional part of the design space?". The comparisons were mostly between single and ensembles for a given number of parameters, but the absolute performance of neither was addressed.
Plots like Figure 2 do not show the absolute accuracy reached, or which part of that plane correspond to "reasonable" performance, which makes it harder to understand if a victory of ensembles in part of that space is important or not. For instance, in the lower-left corner of plots of Figure 3, one could think that the better performance of ensemble methods does not matter much since it is before the model has converged, and that the performance of all models would be low.
As another example, figure 4 shows that for larger models, ensembles tend to reach the accuracy of a single *comparable* model, and do so faster, but that would not mean much if the performance of that single model was bad for some reason (overfitting, for instance).
It is only in the Appendix (Figure A) that we see that the performance of ensemble actually gets better than the *best* single model, rather than only the comparable one. I think that figure deserves to be in the main paper, and would help make the argument.

Additional feedback
----------------------------
- In Figure 1 (b) and (c), colors seem to be mixed up between E2 and E3.
- Duplicate reference to Lee et. al (2015): a and b are the same article.

---

> ### Author Response · Authors · 2020-11-21
> **Experiments on downsampled ImageNet added | Code released | Presentation revised**
>
> Thank you so much for the thoughtful feedback. We reply to all of the reviewer’s questions and revise the paper to include more experiments as pointed out by the reviewer.
>
> **Code**
>
> We have cleaned up and released the code. We now include an anonymous link to the code in the paper at the end of Section 1.
>
> **Large datasets**
>
> We agree with the reviewer that our data set sizes are in the small to medium range. This is an artifact of the fact that due to the detailed design space we are trying to cover, experiments require a significant amount of time.
>
> To verify our observations on a larger data set within the time constraints of the review period, we run our experiments on a downsampled version of ImageNet-1k, where every image is downsampled to be of size 32 x 32 [1]. This data set has 10 times more images than tiny-imagenet, the largest data set in the initial submission, and twice as many labels.
>
> The results of this experiment can be seen in Figure I in the appendix. Overall, our observations hold for this larger data set as well. After a certain parameter budget (defined in terms of number of parameters and training epochs), ensembles of neural networks consistently outperform the single network and also reach that accuracy faster when compared to the single model. We included 4 different parameter budgets in the revised version of the paper in Figure I. We expect to have results for two more parameter budgets by the end of the review period and will update the paper accordingly.
>
> We will also continue to extend this study by including observations from progressively more complex data sets including ImageNet-21k (that for the exhaustive methodology we use will take several months to complete). The website where we host the visualization of the complete set of results will serve as a way of disseminating this information as it becomes available.
>
> As far as model sizes are concerned, we would like to point out that we experiment with a wide range of model sizes: For every architecture, the range of model sizes we consider span the full range of model sizes mentioned in the respective papers where these models were introduced.
>
> Thank you again for this remark.
>
> [1] A Downsampled Variant of ImageNet as an Alternative to CIFAR datasets (https://arxiv.org/abs/1707.08819)
>
> **Absolute performance graphs in the main paper**
>
> Thank you for this excellent suggestion. We agree with the reviewer that including final performance graphs in the main paper provides a more holistic picture as well as underscores the advantage ensembles have over single networks for this important metric, further highlighting the primary insights we want to bring forward. We utilize the additional page allowed during the rebuttal period and moved Figure A (a through c) from the appendix of the initial submission to the main paper. This is now Figure 3 (a through c) in the revision submission.
>
> **Minor**
>
> We addressed all minor comments provided by the reviewer in the latest version of the paper. Thank you so much for bringing them to our attention.
>
> **.: Changes to the paper :.**
>
> (1) Added Figure I in the appendix: Experiments on larger data set with ImageNet32
>
> (2) Figure A (a through c) is now in the main paper as Figure 3 (a through c)
>
> (3) Figure 1 color mix-up is now fixed
>
> (4) Lee et al. duplicate references are now fixed
>
> (5) Added anonymous link to the code at the end of section 1

---

> > ### Author Response · Authors · 2020-11-25
> > **Two new parameter budgets for ImageNet-32 added**
> >
> > We have now added results from two more parameter budgets for DenseNets trained on ImageNet32 (3.7M and 6.11M). This can be seen in Figure I in the appendix. The overall trends in accuracy as well as resource-related metrics hold consistently for this larger data set as well.
> >
> > We thank the reviewer again for their constructive feedback.

---

### Public Comment · ~Ekaterina_Lobacheva1 · 2020-11-16
**On the novelty of the work and missing references**

We enjoyed reading the paper, however, we would like to kindly note that the EST phenomenon described in the paper has been already extensively investigated in several independent works.

Particularly, the effect of a single large network performing worse than a deep ensemble of several smaller networks with the same total budget has been firstly noticed in (Dutt18) and deeply investigated in (Chirkova20) and (Kondratyuk20), both papers uploaded to arXiv in spring 2020. (Chirkova20) investigate the effect for WideResNet and VGG on CIFAR datasets and Transformer on IWSLT14 German-English, finding optimal hyperparameters for the network of different sizes (widths). This paper also analyses the optimal number of networks in the ensemble. The follow-up (Lobacheva20) also proposes a way of predicting the optimal number of networks in the ensemble. (Kondratyuk20) notice the effect for WideResNets on CIFAR and EfficientNet on ImageNet, measuring budgets in FLOPs and varying the width of the networks. This paper also proposes a Neural Architecture Search approach to finding the optimal configuration of the ensemble.

(Dutt18) Dutt, A., Pellerin, D., and Quénot, G. Coupled ensembles of neural networks. International Conference on Content-Based Multimedia Indexing (CBMI), 2018. \
(Kondratyuk20) Kondratyuk, D., Tan, M., Brown, M., and Gong, B. When ensembling smaller models is more efficient than single large models. Arxiv 2020 \
(Chirkova20) Chirkova, N., Lobacheva, E., Vetrov, D. Deep Ensembles on a Fixed Memory Budget: One Wide Network or Several Thinner Ones? Arxiv 2020 \
(Lobacheva20) Chirkova, N., Lobacheva, E., Kodryan M., Vetrov, D. On power laws in Deep Ensembles. NeurIPS 2020

---

> ### Comment · AnonReviewer3 · 2020-11-20
> **On the novelty part raised**
>
> Dear authors, Can you please comment on the novelty part and the missing citations raised by Ekaterina?

---

> > ### Author Response · Authors · 2020-11-21
> > **Please see the response above**
> >
> > We kindly direct the reviewer to our response to the public review above. Thank you.

---

> ### Author Response · Authors · 2020-11-21
> **EST is broader | EST is one of several observations in the paper | We consider a significantly larger design space**
>
> Thank you so much for bringing recent related work to our attention. We cite and address this line of work in the revised version of the paper.
>
> We would like to point out that our study brings a significant amount of additional novelty and depth to both methodology and observations regarding the question of ensembles vs. single networks:
>
> (i) The EST as presented in our work is broader than the MSA explored in past work as it considers six different metrics instead of only accuracy.
>
> (ii) In fact EST is only one of a total of five critical observations we present in this paper as enabled by our holistic analysis framework.
>
> (iii) We apply this holistic framework and verify the observations to a significantly larger design space of network architectures, configurations, and data sets than has been done by all of the above cited papers combined.
>
> Below we elaborate on these three points.
>
> **1 - EST is broader than MSA**
>
> The cited papers define the limit beyond which ensembles dominate only w.r.t the size of the model (defined in terms of FLOPs in [Kondratyuk20] and relative to a standard parameter budget in [Dutt18], [Lobacheva20] and [Chirkova20]).  [Lobacheva20] and [Chirkova20] refer to the threshold beyond which ensembles dominate as the MSA (Memory Split Advantage). EST is a broader concept: It is defined w.r.t both the number of parameters as well as the number of training epochs. This fact is clearly mentioned right in the introduction when we first introduce EST. Considering the size as well as the training budget of the models is crucial to having a holistic analysis and enables us to make much more observations: For example, we show that not only do ensembles provide better accuracy after the EST but can also reach that accuracy faster than a single model. This observation is a direct outcome of the broader definition of the EST where we consider the relative ranking of single and ensemble designs with respect to the number of training epochs and not just the model size.
>
> **2 - Holistic set of metrics -- diverse set of observations**
>
> Our analysis framework considers a holistic set of metrics: 1) accuracy, 2) training time per epoch, 3) total training time, 4) time to accuracy, 5) inference time, and 6) memory usage. All papers mentioned by the public reviewer consider only a single metric, accuracy, under a fixed memory budget.
>
> Our holistic approach reveals several critical observations that are not just about switching based on accuracy: (i) Ensembles can even train to a higher accuracy faster, (ii) Weq ensembles can infer as fast as the single network, (iii) Ensembles require significantly less maximum memory to train as compared to single networks, (iv) Deq are more accurate than weq ensembles while weq can train and infer faster.
>
> **3 - More extensive design space, architectures, and data sets**
>
> The cited papers cover a limited part of the possible design space: in conjunction all papers together consider only 2 data sets (CIFAR and ImageNet) and 3 convolutional architectures (VGGNets, WideResNets, and EfficientNet). For each architecture, these papers consider only a single depth (e.g., [Lobacheva20] considers only WideResNet-28 and VGG-16; to sweep the parameter budget, they vary the width of these singular designs only).
>
> In fact, increasing the width of a network in isolation has already been shown to be a far less effective method to improve the accuracy of VGGNets ([Dauphin13], [Eigen13], and [Ba14]). Since [Lobacheva20] and [Chirkova20] only vary the width of the single network, these comparisons can be  biased in favor of ensembles.
>
> In contrast, our work, in a single experimental framework, considers 3 different convolutional architectures and 5 different data sets, while for every dataset-architecture pair we sweep both the width and the depth of the single network. Corresponding to every width and depth, we create ensembles belonging to two different design classes (width-equivalent and depth-equivalent ensembles). In addition, all this analysis is done across 6 different metrics instead of 1. Overall, we present over 150 comparisons (amongst Single networks, weq, and deq) across all six metrics.
>
> Our study, in this way, provides the most holistic answer to the question of when and how to use ensembles of convolutional neural networks over single network models. In a new paragraph at the end of the paper we now also point out limitations of our study and future steps for the community to continue this analysis.
>
> **References**
>
> [Dauphin13] Yann et al. Big neural networks waste capacity. (2013).
>
> [Eigen13] Eigen et al. Understanding deep architectures using a recursive convolutional network. (2013).
>
> [Ba14] Ba et al. Do deep nets really need to be deep?  (2014).
>
> **.: Changes to the paper :.**
>
> (1) 2nd paragraph page 2: We acknowledge, cite, and position against the new related work.
>
> (2) End of section 4: New paragraph on ‘EST vs. Memory Split Advantage’.

---

### Decision · Program_Chairs · 2021-01-07
**Final Decision**

**Decision:**

Accept (Poster)

**Comment:**

The paper looks into performance of a single network vs ensemble CNN networks of similar no. of parameters, through lens of accuracy, training time, memory used, inference time.
the authors show that after some threshold, the ensemble model starts to outperform a single model and make better use of its capacity.
although this is not the first paper to look into this question and there are two other earlier results from this year, the current paper looks into more measures and not just accuracy.
Although initially the paper only looked at over-parameterized regime, the authors added experiments on under-parametrized case as well. moreover, the authors address the issue of only looking into small and medium sized datasets by adding two more ImageNet experiments.

I thank the authors for engaging with the reviewers, addressing their comments and updating the paper accordingly.

It's of interest for follow up work to consider large data regime and transformer style models as well.